# Effects of a Workplace-Based Virtual-Run Intervention Among University Employees

**DOI:** 10.3390/ijerph17082745

**Published:** 2020-04-16

**Authors:** Apichai Wattanapisit, Waluka Amaek, Watcharawat Promma, Phatcharawadee Srirug, Uchane Cheangsan, Satit Khwanchum, Wattana Chadakorn, Kanittha Eardmak, Narumon Chadakorn

**Affiliations:** 1School of Medicine, Walailak University, Thasala, Nakhon Si Thammarat 80161, Thailand; 2Walailak University Hospital, Thasala, Nakhon Si Thammarat 80161, Thailand; 3Walailak University Running for Health Club, Thasala, Nakhon Si Thammarat 80161, Thailand; 4College of Graduate Studies, Walailak University, Thasala, Nakhon Si Thammarat 80161, Thailand; 5School of Management, Walailak University, Thasala, Nakhon Si Thammarat 80161, Thailand; 6School of Allied Health Sciences, Walailak University, Thasala, Nakhon Si Thammarat 80161, Thailand; 7School of Political Science and Laws, Walailak University, Thasala, Nakhon Si Thammarat, 80161 Thailand; 8Center for Digital Technology, Walailak University, Thasala, Nakhon Si Thammarat 80161, Thailand; 9Center for Scientific and Technological Equipment, Walailak University, Thasala, Nakhon Si Thammarat 80161, Thailand; 10Center for Academic Services, Walailak University, Thasala, Nakhon Si Thammarat 80161, Thailand; 11Division of Finance and Accounting, Walailak University, Thasala, Nakhon Si Thammarat 80161, Thailand

**Keywords:** physical activity, running, virtual run, walking, workplace

## Abstract

Virtual runs (real running or walking activities using online recording platforms) have been popular in the digital age and could have the potential to promote physical activity (PA) in workplaces. We investigated the effects of a virtual-run intervention (VRI) on PA and body compositions among university employees. A three-phase intervention was conducted over 6 months: 0 (pre-intervention), 1 (during intervention), and 2 (post-intervention). Exercise stages of change were assessed in phases 0 and 2. Body compositions (body weight, body mass index, fat mass, percentage of fat mass, fat-free mass, and percentage of fat-free mass) were assessed in phases 0, 1, and 2. Running and walking times were recorded through a mobile application. Forty-seven participants completed the VRI. The number of participants at the maintenance stage increased from 34.04% in phase 0 (*n* = 16) to 63.83% in phase 2 (*n* = 30). None of the changes in body compositions were significant (*p* > 0.05). The median running and walking time among participants at the maintenance stage was 151.85 (interquartile range, 109.15) min/week. Future research should focus on approaches to improve the efficacy of VRIs and their effects on health outcomes.

## 1. Introduction

Physical inactivity or insufficient physical activity (PA) is a global public health challenge [1,2]. Worldwide, more than one-fifth of adults are insufficiently physically active [1]. One possible reason for physical inactivity is a busy lifestyle and heavy workload [3,4]. White collar and professional workers spend a large proportion of their working time on light-intensity PA and sedentary behaviors [5,6]. Specifically, university employees are likely to be sedentary, physically inactive, and have a high prevalence of high body mass index (BMI) and blood pressure [7]. Fountaine et al. investigated occupational sitting and PA among four categories of university employees (administration, faculty, staff, facilities management) [8]. They showed that most university employees participated in low levels of leisure-time PA and spent ~75% of a workday on sitting activities (except facilities-management workers) [8]. 

Promoting PA in workplaces is a potential approach to increase PA and improve health outcomes among employees [9,10,11]. Moreover, a systematic review by Grimani and colleagues showed that health-promotion interventions in the workplace, including PA and nutrition, can improve productivity, work performance, and workability [12]. Intervention patterns to promote PA in workplaces are diverse [13]. In a context of a university worksite, Butler and colleagues presented the positive effects of an intervention on daily step counts and cardiovascular health outcomes [14]. Promoting PA among employees in university worksites requires the understanding of specific barriers to participation in health-promotion programs [15].

Virtual runs or virtual races have become popular in recent years [16]. Each virtual run consists of different components and characteristics [17,18]. A virtual run has not been defined clearly. In general, a virtual run is defined as a running or walking activity using an online platform to record activities. Specifically, a virtual run needs a real running or walking activity (a real activity) and an online recording platform (a virtual platform). Participants use a time-based goal (e.g., run or walk for 30 days) or a distance-based goal (e.g., run or walk for 100 km). 

The possible advantages of virtual runs are flexibility (participants can run or walk at any time and place) and the recording systems. This setup can help participants adjust their PA strategies and achieve PA goals. However, there is a knowledge gap of the effects of virtual runs on PA and health outcomes. We investigated the effects of a workplace-based virtual-run intervention (VRI) on PA and body compositions among university employees.

## 2. Materials and Methods 

### 2.1. Study Design 

A pre- and post-intervention study was conducted between March and October 2019. The study consisted of three phases: (i) phase 0 (pre-intervention); (ii) phase 1 (during intervention); (iii) phase 2 (post-intervention). 

### 2.2. Setting and Participants 

The workplace-based study was conducted on a campus at Walailak University (Nakhon Si Thammarat, Thailand). The total number of university employees recruited was 1213. These university employees were invited to participate in the ‘virtual run program’. Exclusion criteria were employees who had contraindications to exercise (e.g., coronary artery disease, uncontrolled hypertension, uncontrolled diabetes mellitus) or core members of the Walailak University Running for Health Club (WURfHC; the authors of this article). The program invitation with a registration link was posted on the intranet webpage of Walailak University and Facebook™ page of the WURfHC.

### 2.3. VRI and Data Collection 

In phase 0 (pre-intervention), participants were requested to complete the demographic and exercise stages of change (ESC) questions via a Google™ Form (Alphabet, Mountain View, CA, USA). Subsequently, participants were assigned to download Endomondo™ (Under Armour, Baltimore, MD, USA) on personal mobile devices. Endomondo is a free mobile application that uses the Global Positioning System to track PA [19]. By the end of March 2019, the core members of the WURfHC (the authors) were assigned, as the group motivators and observers, into five running groups. Each running group consisted of a combination of participants with mixed ESC (pre-contemplation, contemplation, preparation, action, maintenance). Each participant could see his/her running and walking time, as well as that of peers, within the group. Within each group, the group motivators also recorded their running and walking times and could encourage participants by text messages and their records in the application. Body compositions (body weight, fat mass, percentage of fat mass, fat-free mass, percentage of fat-free mass) were measured by bioelectrical impedance analysis using a Tanita model SC330P machine (Tanita, Tokyo, Japan). 

In phase 1 (during intervention), participants had undertaken a virtual run (running and walking activities at any location and any time) over a 6-month period during the summer to early rainy season, which was a suitable period for outdoor activities (April to September 2019). Participants were requested to record only running and walking activities for travelling and recreational reasons. The participants could record their running and walking activities through three methods: (i) carrying a mobile device and turning on the application during activities; (ii) recording the activities using wearable devices (e.g., smart watches) and transferring the records to the application; (iii) recording the activity time on the application manually after activities. Participants were asked to omit recording occupational PA. In July 2019 (at 3 months of intervention), the body compositions of participants were measured. 

In phase 2 (post-intervention), participants answered a questionnaire regarding ESC. Body compositions were measured. Accumulation of running and walking time was collected from the application by the group observers. Running jerseys were rewarded for participants who participated in more than 600 min of running and walking activities within 6 months. Table 1 shows the summary of the interventions. 

### 2.4. Outcome Measurement and Data Analyses 

#### 2.4.1. Demographic Data and ESC

Age of participants was described as the mean ± standard deviation (SD). Sex was presented as frequencies and percentages. The ESC questionnaire consisted of four dichotomous questions (Yes or No): (i) ‘I am currently physically active (at least 30 min per week)’; (ii) ‘I intend to become more physically active in the next 6 months’; (iii) ‘I currently engage in regular physical activity’; (iv) ‘I have been regularly physically active for the past 6 months’ [20]. The interpretation was a stage of change (Table 2) [20,21]. 

The ESC of each participant was described and compared between phase 0 and phase 2. Changes in the ESC were translated to ‘improved’ (higher stage), ‘remained’ (similar stage), or ‘declined’ (lower stage). 

#### 2.4.2. Statistical Analysis

Body weight, fat mass, and fat-free mass were measured in kilograms. Percentage of fat mass and fat-free mass were recorded. BMI was calculated automatically by the measuring machine. Body compositions were presented as the mean ± SD or median and interquartile range (IQR), as appropriate, in phase 0, phase 1, and phase 2. Each variable of body composition was tested for a normal distribution using the Shapiro–Wilk test. Trends (three-point analysis) of body compositions were evaluated using one-way analysis of variance (one-way ANOVA) or the Kruskal–Wallis test. The total running and walking time was presented in minutes. The weekly time was calculated by dividing the total time with the number of weeks of participation in the intervention (total time in minutes/26 weeks). The median time spent on running and walking and IQR were calculated and classified into each ESC. A value of *p* < 0.05 was considered significant, and data were analyzed using GraphPad Prism v8.3.1 (GraphPad, San Diego, CA, USA).

### 2.5. Ethical Approval of the Study Protocol

The study protocol was approved by the Human Research Ethics Committee of Walailak University (WUEC-19-034-01). Participation in the study was entirely voluntary. Details of the study were provided on the first page of the online questionnaire in phase 0. Participants were asked to provide their consent by ticking a checkbox before taking part in the study. 

## 3. Results

### 3.1. Participants 

A total of 103 employees (64 females and 39 males; mean age, 37.82 ± 9.15 years) completed the questionnaire in phase 0. Fourteen employees did not participate in the VRI. Therefore, 89 participants (53 females and 36 males; mean age 37.37 ± 9.22 years) started the VRI. Forty-seven participants (30 females and 17 males; mean age 39.94 ± 9.12 years) completed the VRI (retention = 52.81%). 

### 3.2. Motivation Towards PA 

The motivations of 103 employees based on the ESC were assessed in phase 0. Forty-two of them (40.78%) were at the maintenance stage. In phase 2, the ESC in 47 participants were evaluated; 30 participants (63.83%) were at the maintenance stage. Of 47 participants who completed the VRI, the improved stage was dominant (*n* = 24, 51.06%). Sixteen were at the remained stage (34.04%) and seven were in the declined stage (14.90%) (Table 3).

### 3.3. Body Compositions 

In phase 0, 47 participants, who engaged in the VRI, completed the assessment for body compositions. In phase 1, 26 participants were assessed. In phase 2, 47 participants attended the assessment for body compositions.

#### 3.3.1. Body Weight and BMI

The median (IQR) of body weight (in kg) was 63.00 (27.00), 65.00 (25.00), and 64.00 (16.00) in phase 0, 1, and 2, respectively. The median (IQR) of BMI was stable from phase 0 to phase 2: 24.00 (4.00) kg/m^2^. The change in body weight (Figure 1a) and BMI (Figure 1b) was not significant (Kruskal–Wallis test).

#### 3.3.2. Fat Mass and Percentage of Fat Mass

Fat mass and percentage of fat mass were not significant over the intervention period (Figure 2). The median (IQR) of fat mass (in kg) was 17.00 (8.00), 17.00 (10.00), and 17.00 (10.00) in phase 0, 1, and 2, respectively (*p* = 0.78 by the Kruskal–Wallis test). The mean ± SD of percentage of fat mass was 27.00% ± 10.00%, 28.00% ± 11.00%, and 27.00% ± 10.00% for phase 0, 1, and 2, respectively (*p* = 0.92 by one-way ANOVA).

#### 3.3.3. Fat-free Mass and Percentage of Fat-free Mass

The change in fat-free mass and percentage of fat-free mass was not significant in phase 0, 1, and 2 (Figure 3). The median (IQR) of fat-free mass (in kg) was 41.00 (15.00), 41.00 (18.00), and 41.00 (13.00) in phase 0, 1, and 2 (*p* = 0.83 by the Kruskal–Wallis test). The mean ± SD of the percentage of fat-free mass was 69.00% ± 9.90%, 68.00% ± 10.00%, and 69.00% ± 9.90% in phase 0, 1, and 2, respectively (*p* = 0.91 by one-way ANOVA).

### 3.4. Running and Walking Time

Table 4 presents the total running and walking time collected in phase 2 and weekly running and walking time (calculated by dividing the total time with 26 weeks). Of 47 participants, the median (IQR) running and walking time was 1545.00 (2290.50) min over 26 weeks of the VRI, which was 59.42 (88.10) min/week. Participants in the maintenance stage (*n* = 30) achieved 151.85 (109.15) min/week. 

## 4. Discussion

The VRI obtained low retention (52.81%) among voluntary participants in a university worksite. Initially, most participants were at the maintenance stage, whereas more than half dropped out of the VRI. In addition, 51.06% of participants who engaged in the entire intervention period reported improvement in the ESC. Body compositions were not changed significantly during the 6-month VRI.

In phase 0, only 8.55% (103/1204; excluding the authors) of university employees were interested in the VRI. Of 103 registered participants, 14 were considered non-usage attrition (never used the intervention) [22]. Of 89 participants who commenced the intervention, 42 dropped out (dropout attrition, 47.19%). Overall, the attrition prevalence was 54.37% (56/103). The attrition prevalence of our study is comparable with that of other digital health interventions in workplaces (range, 0% to 60%; median, 21%) [23]. This intervention offered some components for which barriers had to be overcome by employees (e.g., using an online platform, offering objective PA measurement), and another approach, such as offering incentives, might improve the retention of study participants [24]. Nevertheless, in terms of promoting PA at workplaces, participatory, multicomponent, and holistic approaches are required [25].

Initially, of 47 participants in the VRI, 16 (34.04%) were at the maintenance stage. At the end of the VRI, 30 (63.83%) were at the maintenance stage. One mechanism that might explain the overall improvement in motivation towards PA was a mixture of participants at different stages in each running group. This approach enabled the Köhler effect among participants: a reaction in a group when a weaker member is motivated by stronger members [26]. Although the overall changes were improved, some participants were stable at similar stages, whereas others regressed to lower stages. Three out of 16 participants relapsed from the maintenance stage to lower stages: this reflected dynamic changes of PA behaviors. A challenge of this VRI was to prevent a relapse of physical inactivity or insufficient PA in active participants.

Although the VRI had the potential to improve motivations towards PA, it could not improve body compositions over the 6-month period. The VRI assigned participants to collect activities time without a specific goal. An ideal virtual run can offer a SMART goal, which can be an effective scheme to enhance physical fitness [27]. The SMART mnemonic refers to: Specific (a specific activity, such as running or walking), Measurable (an objective recording system), Achievable (a program for different individuals), Results-focused (a realistic result), and Timely (a realistic timeframe) [28]. Virtual-run administrators should manage a clear and specific goal for each VRI.

Overall, participants spent 59.42 min/week on running and walking activities through the intervention period. Participants at the maintenance stage (151.85 min/week) achieved the weekly recommended time, but PA intensity could not be measured. Current PA recommendations suggest that adults should undertake ≥150 min/week of moderate-intensity aerobic PA or ≥75 min/week of vigorous-intensity aerobic PA or an equivalent combination of moderate-to-vigorous-intensity PA [29,30,31]. In addition, a virtual run should set the moderate-to-vigorous-intensity PA as a threshold. This may be a key factor to improve VRI efficacy.

Our study had three main strengths. First, the VRI offered an easy and convenient program in a workplace-based setting in which participants could engage at any time and place. Second, the PA recording system was an objective measurement that provided several simple methods to record running and walking activities. This approach could support participants with various mobile devices. Although one method was a manual recording method, running and walking time was easily recorded and validated. Third, a long intervention period allowed observation of changes in PA motivations over the 6-month period.

The study had three major limitations. First, the study cohort was small. This reflected a lack of interest in the VRI among university employees. Second, the prevalence of attrition was high. A possible reason was that the motivational incentives might not be enough to retain the number of participants. Third, PA records did not indicate PA intensity. For example, a participant recorded a certain time for a walking session, but this information did not specify the intensity of the activity.

## 5. Conclusions

The VRI could improve motivations towards PA among university employees. However, some challenges (high prevalence of attrition, a non-improvement of body composition, non-achievement of PA recommendations) were addressed. Future research should focus on approaches to improve the efficacy of VRIs and their effects on health outcomes.

## Figures and Tables

**Figure 1 ijerph-17-02745-f001:**
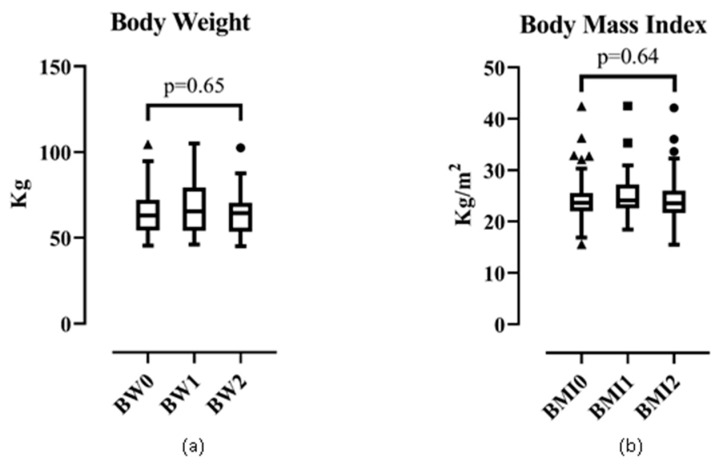
Trend of (**a**) body weight and (**b**) body mass index.

**Figure 2 ijerph-17-02745-f002:**
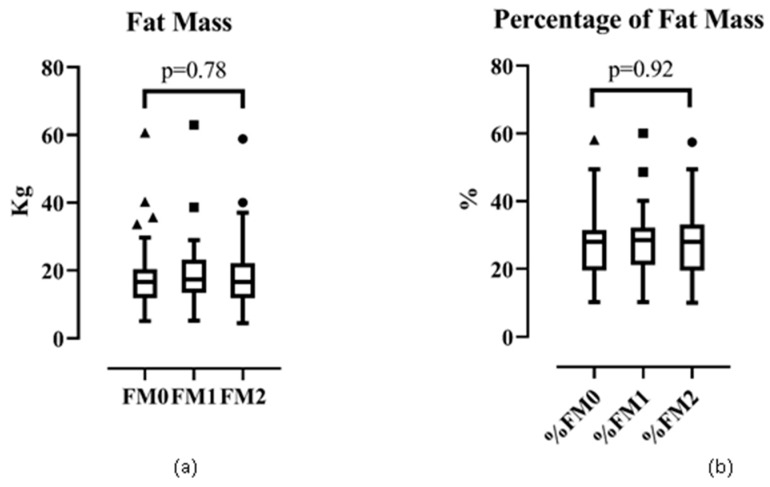
Trend of (**a**) fat mass and (**b**) percentage of fat mass.

**Figure 3 ijerph-17-02745-f003:**
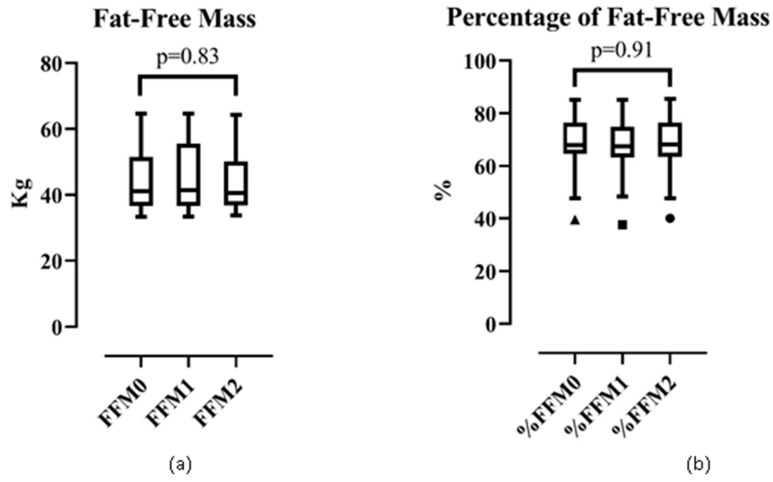
Trend analysis of: (**a**) fat-free mass and (**b**) percentage of fat-free mass.

**Table 1 ijerph-17-02745-t001:** Summary of virtual-run intervention and data-collection processes.

Phase 0 (Pre-Intervention)	Phase 1 (During Intervention)	Phase 2 (Post-Intervention)
• **Demographic data—**collecting data via a Google™ Form • **1st exercise stages of change****—**categorizing into five stages (pre-contemplation, contemplation, preparation, action, maintenance) • **1st measurement of body compositions****—**including body weight, fat mass, percentage of fat mass, fat-free mass, percentage of fat-free mass • **Download of mobile application—**a free mobile application (Endomondo™) • **Group assignment****—**each group consisted of 1–2 group motivators and observers (WURfHC members) and a combination of participants with mixed exercise stages of change	• **Running and walking activities** (six months)—all participants in the five groups were voluntary to participate in running and walking activities without different assigned interventions • **Recording running and walking times****—**participants recorded each running or walking activity through Endomondo™ • **2nd measurement of body compositions** (at 3 months of intervention)	• **2nd exercise stages of change** • **3rd measurement of body compositions** • **Collecting running and walking times****—**the group observers collected the overall running and walking time of each participant • **Rewarding—**running jerseys for >600 min participation in running and walking activities

**Table 2 ijerph-17-02745-t002:** Interpretation and description of exercise stages of change.

Exercise Stages of Change	Interpretation of the Questionnaire *	Description
Precontemplation stage	No to questions (i), (ii), (iii), and (iv)	No intention to take action in the next 6 months
Contemplation stage	No to questions (i), (iii), and (iv) Yes to question (ii)	Intention to change in the next 6 months
Preparation stage	Yes to questions (i) and (ii) No to questions (iii) and (iv)	Intention to take action in the immediate future
Action stage	Yes to questions (i) and (iii) Yes or No to questions (ii) No to question (iv)	Current action within 6 months
Maintenance stage	Yes to questions (i), (iii), and (iv) Yes or No to question (ii)	Maintenance of the action more than 6 months

* (i) ‘I am currently physically active (at least 30 min per week)’; (ii) ‘I intend to become more physically active in the next 6 months’; (iii) ‘I currently engage in regular physical activity’; (iv) ‘I have been regularly physically active for the past 6 months’.

**Table 3 ijerph-17-02745-t003:** Exercise stages of change in phase 0 (pre-intervention) and phase 2 (post-intervention).

	**Exercise stages** **of change**	**Phase 2**						
		Precontemplation	Contemplation	Preparation	Action	Maintenance	Dropout *	Total (phase 0)
**Phase 0**	Precontemplation					1 ^+^	1	2
	Contemplation			4 ^+^	1 ^+^	2 ^+^	9	16
	Preparation		1^–^	1 ^0^	2 ^+^	6 ^+^	9	19
	Action	1^–^	1^–^	1^–^	2 ^0^	8 ^+^	11	24
	Maintenance		1^–^	1^–^	1^–^	13 ^0^	26	42
	Total (phase 2)	1	3	7	6	30	56	103

* Of 56 dropouts, 14 did not start the intervention (non-usage attrition), and 42 were lost to follow-up (dropout attrition). ^+^ Number of improved stages. ^0^ Number of remained stages. ^–^Number of declined stages.

**Table 4 ijerph-17-02745-t004:** Running and walking time by exercise stages of change.

Exercise Stages of Change in Phase 2	Total Running and Walking Time (26 Weeks) Median (IQR) (Min)	Weekly Running and Walking Time Median (IQR) (Min/Week)
Precontemplation (*n* = 1)	150.00 (0.00)	5.77 (0.00)
Contemplation (*n* = 3)	48.00 (78.50)	1.85 (3.02)
Preparation (*n* = 7)	1616.00 (1154.50)	62.15 (44.40)
Action (*n* = 6)	925.00 (1296.00)	35.58 (49.85)
Maintenance (*n* = 30)	3948.00 (2838.00)	151.85 (109.15)
Total (*n* = 47)	1545.00 (2290.50)	59.42 (88.10)

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
