# Peer review of "Effects of a Workplace-Based Virtual-Run Intervention Among University Employees"

_ijerph, 2020, doi:10.3390/ijerph17082745_

Round 1
Reviewer 1 Report
This manuscript studied the effects of the workplace-based virtual-run intervention on physical activity and body compositions among the university employees. Overall the manuscript is well written. The background describes the clear rationale, the study aims are clear, and the methods part is mostly clear.
However, I am not very clear how this study is a workplace-based intervention? The intervention (what was provided) is not sufficiently described. Also, clarify whether all five groups got the same intervention or different and if that was provided at the workplace.
The five-exercise stage of change is not sufficiently described in the manuscript.
Please clarify whether the mobile application provided data on running and walking time only or more detail such as energy consumption to know the level of physical activities the participants did. Also, it is not clear whether the researcher had direct access to the data on the physical activities of the participants or not. Overall, the validity of the information obtained from the mobile application should be described.
Currently, there is no statistical part included in the manuscript which makes it difficult to understand the results.
A study flow diagram of the participants should be provided.
The results are not sufficiently presented and described, for e.g. Table 3 is not clear what authors are trying to show. I was expecting the difference in the pre-test and post-test in the physical activity among the participants.
In the discussion, the study strengths and limitations are not sufficiently discussed for e.g. the high attrition is the major limitation of the study, but it is not well addressed in the manuscript.
In conclusion, please revise the first sentence, currently ‘the VRI conducted at our university ……’ does not sound good.
Â
Author Response
Thank you for your review and helpful comments.Â
We have responded to all comments. Please see the attached file.

Reviewer 2 Report
I found the study significant in this day and age as Unveristy across nations and global are encouraging I compliment the authors for their honesty and insights to strengths and limitations of the study especially regarding attrition and lack of data on exercise intensity. This is a plus to anyone looking to duplicate the study. I find the study very significant because across college campuses are program encouraging employees to engage in physical activity. The research design is sound with results and analysis of results. A couple of concerns that may need to be explained.
- The length of the study could of contributed to the attrition
- explain why April to September the study was conducted? in the United States this would be the time of year University employees would be using vacation time or only 9 month employees. So in Thailand would this be the time of year the University is in session? That needs to be explained.Â
- What incentives might of been provided to encourage employees to participate and to stay in the study. e.g. were there incentives like a raffles, t-shirts, prizes for those that reach each stage of the study. e.g we have e.g 8 week Spring semester employee program similar run, walk, swim, bike virtual programs. Those that complete the 8 weeks, hours names go in a raffle. other programs everyone gets a t-shirt..etc
- our University employee exercise programs are also encouraged by our University health insurance provider. Not sure if your university has employee health insurance policy. The more employees that have a good health report each year the better our insurance rates for all. So their is an incentive from insurance rates. Again not sure if you have that as an incentive to decrease the attrition rate.
- Did you consider employees to partner with another employee to motivate each other to complete the program?
Author Response

(The authors gave the same response as above.)

Round 2
Reviewer 1 Report
I appreciate the hard work of authors in revising the manuscript. Most of the queries were answered in the revised manuscript. However, one of my suggestions to provide 'statistical analysis' in its own heading was not addressed. Currently, the statistical analysis technique used is described under section '2.4.2Â Body Compositions and Running and Walking Time'. For clarity and, quality of the manuscript, I recommend that the 'statistical analysis' should stand its own.
Author Response
Reviewer's comment:Â
I appreciate the hard work of authors in revising the manuscript. Most of the queries were answered in the revised manuscript. However, one of my suggestions to provide 'statistical analysis' in its own heading was not addressed. Currently, the statistical analysis technique used is described under section '2.4.2Â Body Compositions and Running and Walking Time'. For clarity and, quality of the manuscript, I recommend that the 'statistical analysis' should stand its own.
Â
Authors' response:Â
Thank you for your kind suggestion. We have revised the heading - '2.4.2. Statistical Analysis'.  Â